# Mitotic spindle scaling during *Xenopus* development by kif2a and importin α

**Jeremy D Wilbur\*, Rebecca Heald\***

Department of Molecular and Cell Biology, University of California, Berkeley, Berkeley, United States

**Abstract** Early development of many animals is characterized by rapid cleavages that dramatically decrease cell size, but how the mitotic spindle adapts to changing cell dimensions is not understood. To identify mechanisms that scale the spindle during *Xenopus laevis* embryogenesis, we established an in vitro system using cytoplasmic extracts prepared from embryos that recapitulates in vivo spindle size differences between stage 3 (4 cells, 37 μm) and stage 8 (~4000 cells, 18 μm). We identified the kinesin-13 kif2a as a driver of developmental spindle scaling whose microtubule-destabilizing activity is inhibited in stage 3 spindles by the transport receptor importin α, and activated in stage 8 when importin α partitions to a membrane pool. Altering spindle size in developing embryos impaired spindle orientation during metaphase, but chromosome segregation remained robust. Thus, spindle size in *Xenopus* development is coupled to cell size through a ratiometric mechanism controlling microtubule destabilization.

## Introduction

Cell size varies widely among different organisms and cell types, and changes rapidly during early animal development when cell division occurs in the absence of growth. The mitotic spindle, a dynamic bipolar structure composed primarily of microtubules (MTs) and organizing proteins (*Walczak and Heald, 2008*), must adjust to cell size to segregate chromosomes the proper distance and signal to the cell cortex to specify orientation of the cleavage furrow (*Levy and Heald, 2012*). However, the mechanisms coordinating cell and spindle size are poorly understood, and the functional consequences of altering these mechanisms during vertebrate development are unknown.

Embryogenesis of the frog *Xenopus laevis* typifies the developmental scaling problem, and both cell and spindle sizes have been measured (*Wuhr et al., 2008*). Following fertilization, the ~1.2-mm-diameter egg synchronously divides 12 times at approximately 30-min intervals, leading to a ~100-fold decrease in cell volume and at least a 20-fold decrease in cell diameter by the midblastula transition (stage 8.5), when zygotic transcription starts. Initially, the cortically localized meiosis II spindle (~35 μm) segregates chromosomes a short distance, generating a small polar body that preserves egg cytoplasm. During the early cleavages, there is an upper limit to spindle size (~60 μm), and astral microtubules mediate movement of chromosomes long distances during anaphase (*Wuhr et al., 2008*). At these stages, spindle size is small compared to cell size, and cytoplasmic mechanisms likely operate, since in vivo spindle size is maintained in extracts prepared from eggs or two-cell stage embryos (*Mitchison et al., 2005*; *Wuhr et al., 2008*; *Loughlin et al., 2011*). As cell and spindle size converge, a constant ratio of ~2:1 cell diameter:spindle length is established, and spindle length reduces to ~20 μm by the end of the synchronous cleavage divisions. To what extent cell size influences spindle size or if changes in cytoplasmic activities contribute to spindle scaling at these later stages is unknown.

Experiment and simulation indicate that the balance of MT nucleation, transport, and destabilization are critical for setting spindle size in egg extracts (*Loughlin et al., 2010*; *Brugues et al., 2012*). By comparing two closely related *Xenopus* species, we showed previously that meiotic spindle size is partially controlled by activity of the MT severing protein katanin (*Loughlin et al., 2011*). However, the

**\*For correspondence:**
jwilbur@berkeley.edu (JDW);
bheald@berkeley.edu (RH)

**Competing interests:** The authors declare that no competing interests exist

**Reviewing editor**: Tony Hyman, Max Planck Institute of Molecular Cell Biology and Genetics, Germany

**eLife digest** In the earliest stages of development, animal cells undergo multiple rounds of division without growth via a process known as mitosis. Over the course of just 12 rounds of cell division, a single fertilized egg is transformed into more than 4000 smaller cells.

Before dividing, the cell must first replicate its chromosomes. A structure called the mitotic spindle then separates the members of each chromosome pair and distributes them evenly between the two daughter cells. Given that the daughter cells become smaller with each round of division, the spindle must also become smaller to ensure that the chromosomes are pulled apart an appropriate distance. However, it has been unclear how the cell achieves this.

Now, Wilbur and Heald report insights into the mechanism by which spindle size is coordinated with cell size, using the model organism *Xenopus laevis*—a frog that produces large embryos that are easy to manipulate. They began by preparing extracts of cytoplasm from *X. laevis* embryos at two different developmental stages: one set of embryos contained 4 cells each and the other set contained ~4000 cells. They found that the spindle, which is composed largely of microtubules— hollow filaments that can become longer or shorter through the addition or removal of tubulin building blocks—was almost twice as large in the four-cell embryos as in the more developed embryos.

Moreover, spindle size was determined by the actions of two proteins: kif2a and importin-α. Binding of kif2a destabilized microtubules and caused them to shorten; importin-α blocked this process by binding to kif2a and preventing it from interacting with microtubules. Wilbur and Heald found that over the course of development, importin-α became increasingly localized to the cell membrane, meaning that there was less available to bind to kif2a in the cytoplasm. This freed up kif2a to interact with and destabilize microtubules, and led ultimately to a reduction in spindle size.

Given that the overall ratio of cell surface membrane to cytoplasm increases as cells undergo division without growth, interaction between kif2a and importin-α could be the long-sought mechanism by which spindle and cell sizes are coordinated early in development.

genetically encoded differences that underpinned variation in katanin activity cannot play a role during early development in an embryo with a single genetic background and when transcription is repressed. In early *Caenorhabditis elegans* development, decreasing spindle length correlates with a decrease in centrosome size and a decay gradient of the protein of TPXL-1 along spindle MTs (**Greenan et al., 2010**). In contrast, spindle assembly in *X. laevis* egg extracts does not require centrosomes and depends heavily on a chromatin-centered gradient of RanGTP, which locally releases spindle assembly factors from the nucleocytoplasmic transport protein importin β and its adaptor importin α that binds nuclear localization sequence (NLS)–containing cargos (**Gruss et al., 2001**; **Kalab and Heald, 2008**). The respective contribution of gradients and centrosomes are expected to alter spindle architecture and size, and could be influenced by cell size through availability of components (**Goehring and Hyman, 2012**). The gradual transition from meiotic to mitotic spindle architecture observed in mouse embryos supports this view (**Courtois et al., 2012**). Collectively, these observations suggest that changes in gradients and spindle assembly pathways are likely to accompany cell size changes and could influence spindle size.

To address how spindle size is determined during vertebrate embryogenesis, we created an embryo extract system capable of assembling spindles in cytoplasmic extracts at any stage of *X. laevis* development up to the midblastula transition when cell divisions become asynchronous (**Newport and Kirschner, 1982**). We compared stage 3 and stage 8, the two extremes of spindle size in our extract system, and found that MT dynamics are modified to reduce spindle size upon release of an inhibitory interaction between the MT depolymerizing kinesin kif2a and importin α. This occurs through redistribution of importin α from the cytoplasm to a membrane fraction. Taking advantage of this mechanism to modify spindle size without functionally perturbing the spindle allowed us to uncover an important role for spindle size in maintaining spindle orientation in the large cells of early *Xenopus* development. This work suggests that the changing physical features of the cell can drive intracellular scaling through biochemical partitioning and steady-state redistribution of activities.

# Results

## Reconstitution of spindle size differences in extracts of developing embryos

To investigate how spindle size scales with cell size during development, we established a spindle reconstitution system utilizing cytoplasmic extracts from *X. laevis* embryos at different embryonic stages. This system is analogous to the cytostatic factor (CSF) arrested meiosis II extract derived from unfertilized *X. laevis* eggs, which has facilitated fundamental discoveries pertaining to the cell cycle and spindle assembly (*Maresca and Heald, 2006*). However, embryo extracts form mitotic rather than meiotic spindles, and spindles assemble to varying sizes corresponding to their developmental stage in vivo (*Figure 1A*). Since embryo extracts lack the natural metaphase arrest induced by CSF activity in egg extracts, it was necessary to artificially arrest them by adding a nondegradable form of cyclin B to maintain high cyclin-dependent kinase 1 activity together with a dominant negative inhibitor of the

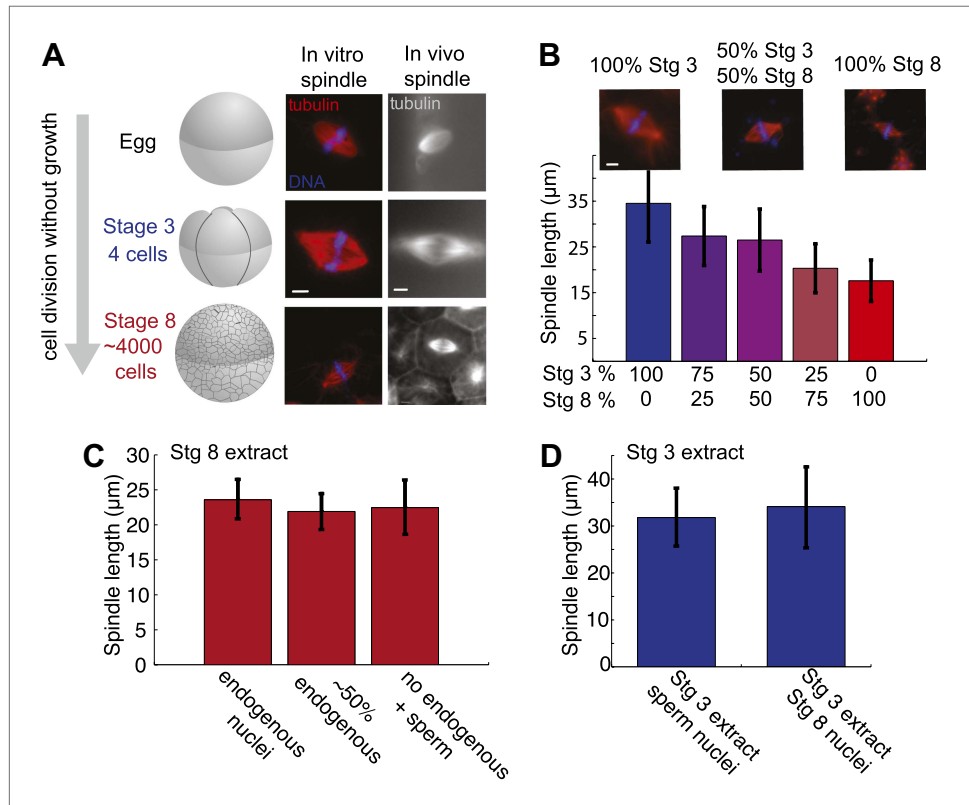

**Figure 1**. Embryo extracts recapitulate spindle scaling during development. (**A**) Schematic of early embryonic developmental stages of *Xenopus laevis* used in this study and representative images of spindles formed in egg and embryo extracts and in fixed hemisected eggs or embryos. Tubulin fluorescence is red or gray and DNA is blue. (**B**) Length analysis of spindles in individual or mixed embryo extracts. Fixed spindles were centrifuged onto coverslips and pole-to-pole length was measured from 49 to 230 spindles in each condition sampled from three independent experiments. Mean ± SD, p<0.0001 by Kruskal–Wallis test. Representative images are shown for mixtures as indicated. The scale bars represent 10 µm. (**C**) Spindles from unfractionated stage 8 extract, stage 8 extract centrifuged at 3000×*g* for 10 min to remove nuclei or stage 8 extract filtered through a 1-µm filter with added demembranated sperm nuclei were sedimented onto coverslips, and pole-to-pole length was measured. Mean ± SD, n ≥ 29 spindles per condition. Differences were not considered statistically significant by an unpaired t-test. (**D**) Spindles were generated in stage 3 extract with nuclei purified from stage 8 extract. Spindles were processed and analyzed as above. Mean ± SD, n ≥ 40 spindles from two extracts. Differences were not statistically significant by an unpaired t-test.

The following figure supplements are available for figure 1:

**Figure supplement 1**. MT flux and severing are similar in embryo extracts.

anaphase promoting complex, UbcH10 C114S (*Rape et al., 2006*). Additionally, an empirically determined time point within each developmental stage was optimal for generating functional extracts. For example, to generate stage 3 extracts from embryos developing at 23°C, packing and crushing was initiated exactly 1 hr and 40 min after fertilization and completed 20 min later, just before the second embryonic cleavage would have occurred.

The morphology of embryo extract spindles resembled that of mitotic spindles in the embryo and frequently possessed astral MTs, unlike meiosis II spindles formed in egg extracts (*Figure 1A,B*). Comparing the extracts that differed most, MTs typically appeared less dense in stage 8 spindles compared to stage 3 or meiosis II spindles, indicating different underlying MT architectures. In vivo spindle size can vary significantly within the different sized cells at a single developmental stage. For instance, stage 8 embryo spindles range from ~20 μm in smaller cells of the animal pole to ~50 μm in length in the large vegetal pole cells. Spindle size varied less within an embryo extract, and on average was slightly reduced compared to the equivalent stage in vivo. Stage 3 populations had average lengths ranging from ~35 to 40 μm and stage 8, ~17 to 20 μm, closer to the size of spindles in animal pole cells. The most common defect observed in embryo extract spindles was the failure of chromosomes to fully align at the metaphase plate, especially in stage 3 extracts, likely due to poor metaphase arrest (unpublished data). These observations indicate that embryo extracts do not entirely preserve spindle morphology and function. However, spindle scaling was apparent in vitro, with a ~20 μm decrease in length between stage 3 and stage 8 extract spindles, compared to ~30 μm in vivo.

To determine whether cytoplasmic mechanisms operate to control spindle size variation in the two extracts, we mixed them in different proportions before initiating spindle assembly. Altering the ratio of stage 3 and stage 8 extracts led to dose-dependent changes in spindle size (*Figure 1B*). Since stage 8 extracts contain abundant endogenous nuclei whose chromosomes are incorporated into forming spindles, we wondered whether the ratio of nuclei to cytoplasm or the contents of endogenous nuclei were affecting spindle size in stage 8 extracts. We therefore compared reactions depleted of endogenous nuclei by centrifugation, or filtered to remove all nuclei, and then supplemented with exogenous sperm nuclei. Average spindle lengths were very similar in the presence of ~2000 nuclei per microliter in untreated stage 8 extract, ~1000 nuclei per microliter in centrifuged stage 8 extract, and ~500 sperm nuclei per microliter in filtered stage 8 extract (*Figure 1C*). Furthermore, purified stage 8 nuclei added to stage 3 extracts yielded indistinguishable spindle lengths compared to sperm nuclei (*Figure 1D*). These experiments indicate that the major spindle size controlling factors in embryo extracts are soluble cytoplasmic components.

## Spindles assembly occurs through different pathways in stage 3 and stage 8 extracts

To distinguish whether spindles assembled through different pathways in the two embryo extracts, we examined spindle assembly intermediates 15 min after reactions were initiated. Whereas MTs nucleated around chromatin in stage 3 extracts, and little interaction of centrosomal MTs with the forming spindle was observed, centrosomes provided most of the MT nucleation in stage 8 extracts and few MTs appeared around the chromosomes except when a centrosome was located nearby (*Figure 2A,B*). These observations suggest that the spindle assembly pathway switches from chromatin-driven in stage 3 to centrosome-driven in stage 8 extracts. To test this idea, we compared the sensitivity of embryo extract spindle assembly to addition of Ran(T24N), a dominant negative inhibitor of the RanGEF RCC1 that destroys the chromatin-induced RanGTP and cargo gradients, severely disrupting spindle assembly in *X. laevis* egg extracts (*Kalab et al., 2002*). Stage 3 spindles were similarly disrupted by this treatment, resulting in a significant (p<0.001, comparing treated to control) loss of tubulin intensity in spindle center, while stage 8 extract spindles were unaffected under the same conditions (p>0.01, *Figure 2C,D*). Thus, stage 3 spindles assemble by a chromatin-driven mechanism requiring a RanGTP gradient, similar to the *Xenopus* meiotic spindle, whereas stage 8 spindle assembly more closely resembles the centrosome-dominated spindle pathway typical of somatic cell types.

## Changes in MT dynamics and spindle recruitment of kif2a correlate with spindle size

We have shown previously that changes in MT stability contribute to spindle length differences in a computational 2D meiotic spindle simulation (*Loughlin et al., 2010*) and between *Xenopus tropicalis* and *Xenopus laevis* egg extracts (*Loughlin et al., 2011*), and hypothesized that altered MT dynamics

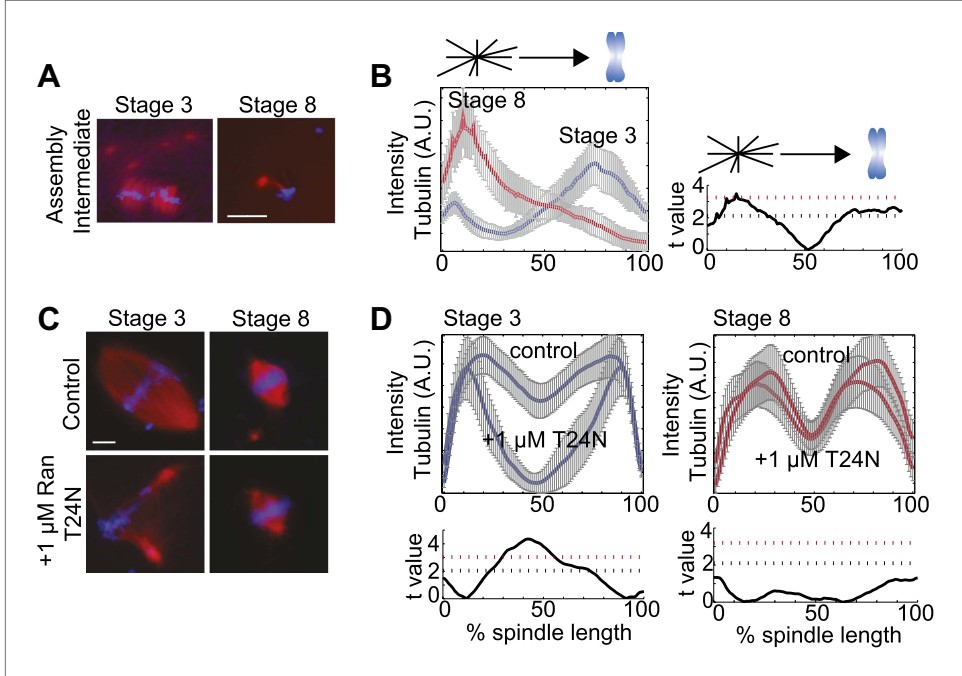

**Figure 2**. Spindle assembly pathways change during development. (**A**) Spindle assembly intermediates 15 min after initiation exhibit different morphologies in stage 3 and stage 8 extracts. (**B**) Left panel: Line scan quantification of tubulin intensity in spindle intermediates starting at the centrosome and ending at chromatin with normalized lengths for statistical analysis. Right panel: t-test statistical analysis across discrete length bins of stage 3 and stage 8 intensities as described in the experimental procedures. Black dashed line is equivalent to a p value of 0.01 and red dashed line is equivalent to a p value of 0.001. (**C**) Inhibition of the RanGTP gradient by addition of 1 μM RanT24N disrupts steady-state spindles in stage 3 but not stage 8 extracts. (**D**) Upper panels: Line scan quantification of tubulin intensity with normalized spindle lengths for statistical analysis. Lower panels: t-test statistical analysis across discrete length bins of control vs 1 μM RanT24N addition as described in the experimental procedures. Black dashed line is equivalent to a p value of 0.01 and red dashed line is equivalent to a p value of 0.001. Mean ± SE from three extracts, tubulin is red and DNA is blue. The scale bars represent 10 μm.

also underlie the different spindle sizes observed in stage 3 vs stage 8 embryo extracts. Therefore, we characterized the dynamic parameters of single MTs nucleated by centrosomes using time-lapse fluorescence microscopy, and found the primary difference between stage 3 and stage 8 extracts to be a threefold increase in MT catastrophe frequency (*Table 1*, *Figure 1—figure supplement 1A*, and *Video 1*), a difference sufficient to reduce spindle length from ~35 to ~25 μm in the meiotic spindle model (*Loughlin et al., 2010*). We measured centrosomal nucleation and MT growth rate by counting and tracking MT plus-ends bound to Alexa 488–labeled EB1 (*Video 2*). Whereas growth rates were similar, a small increase in MT nucleation corrected for average rescue frequency was documented in stage 3 extracts, but whether this is relevant to nucleation in the spindle is unclear. The rate of poleward MT flux determined by fluorescent speckle microscopy was similar in stage 3 and stage 8 spindles and comparable to that of meiotic spindles (*Table 1* and *Figure 1—figure supplement 1B*; *Maddox et al., 2003*; *Brown et al., 2007*; *Ohi et al., 2007*). Finally, MT severing assayed with fluorescent, taxol-stabilized MTs was also very similar in stage 3 and stage 8 extracts (*Figure 1—figure supplement 1C*). These results suggest that an increased MT catastrophe frequency primarily drives developmental spindle scaling.

To determine if the increased catastrophe frequency found in stage 8 extracts correlated with changes in the known catastrophe-causing enzymes, we examined spindle recruitment of the two kinesin-13 family members found in *Xenopus*, kif2a, and MCAK. Quantification of immunofluorescence intensity revealed that compared to stage 3 spindles in which kif2a localized weakly to kinetochores and spindle poles, increased amounts of kif2a homogeneously localized to stage 8 extract spindles, with a slight enrichment at spindle poles (*Figure 3A,B*). The case was markedly different for MCAK,

**Table 1.** Microtubule dynamics and spindle MT flux in embryo extracts

| | Stage 3 extract | Stage 8 extract | Egg extract |
|---|---|---|---|
| MT growth rate—EB1 tracking (µm/min) | 20.6 ± 2.9 | 18.9 ± 2.4 | 12.4 ± 2.4 (*11.5–18.5*) |
| Nucleation frequency (min⁻¹) | 7.0 ± 1.2 | 5.1 ± 0.78 | 8.3 ± 0.36 |
| Catastrophe frequency** (min⁻¹) | 0.54 ± 0.3 | 1.8 ± 0.54 | 0.72 ± 0.54 (*0.9–5.82*) |
| Rescue frequency* (min⁻¹) | 1.08 ± 0.96 | 0.48 ± 0.3 | 0.78 ± 0.3 (*0.004–0.05*) |
| Spindle MT flux rate (µm/min) | 2.38 ± 0.33 | 2.17 ± 0.28 | N.D. (*1.79–2.3*) |

*p<0.01, **p<0.001 between stage 3 and stage 8 extracts by t-test.
Values in italics indicate the range of previously made measurements in egg extracts.
References for MT dynamics in egg extracts (**Wilde et al., 2001**; **Tournebize et al., 1997**; **Carazo-Salas et al., 2001**; **Maddox et al. 2003**; **Tirnauer et al., 2004**; **Brown et al., 2007**). N ≥ 30 microtubules in at least 3 extracts for dynamics and three to five spindles in three extracts for flux.

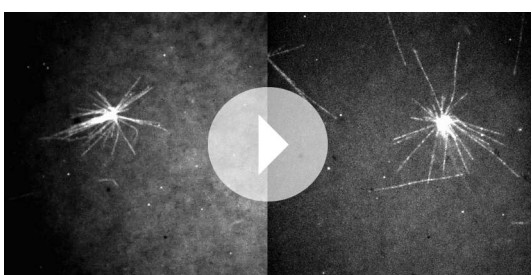

**Video 1**. Centrosomal MT dynamics in embryo extracts. Purified human centrosomes were added to either stage 3 or stage 8 extracts with Alexa 488–labeled tubulin. After ~5-min equilibration, time-lapse images were collected every 5 s for 2 min.

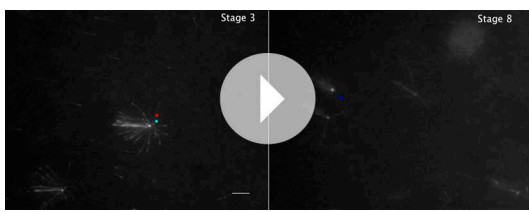

**Video 2**. EB1 comets on MT plus ends emanating from centrosomes in stage 3 and stage 8 extracts. Purified human centrosomes were added to either stage 3 or stage 8 extracts together with Alexa 488–labeled EB1. Time-lapse images were collected every 2 s for up to 1 min. Examples of tracking are shown for four EB1 comets.

which displayed a similar localization in both spindle types and had higher overall intensity in stage 3 extract spindles, indicating that it is unlikely to be involved in developmental spindle scaling. Changes in kif2a spindle recruitment between stage 3 and stage 8 spindles occurred without significant changes in protein levels (*Figure 3C*). Many spindle components implicated in regulation of catastrophe or nucleation were quantitatively similar in localization between stage 3 and stage 8 extracts. These include the MT minus-end protector patronin, the MT nucleator γ-tubulin, and the mitotic kinase aurora B (*Figure 3—figure supplement 1A–C*). Other spindle proteins including TPX2 and the kinetochore kinesin CENP-E were qualitatively similar and likewise were not selected for further analysis (*Figure 3—figure supplement 1D*). Thus, kif2a appeared to be the best candidate for a cytoplasmic developmental spindle scaling factor.

If kif2a functions in developmental spindle scaling, we predicted that its inhibition would differentially affect spindle size in the two extracts. Inhibitory anti-kif2a antibodies added to spindle assembly reactions had no effect on steady-state spindle morphology or size in stage 3 extracts (*Figure 3D*), but caused a dramatic increase in spindle size and change in spindle morphology to a more round lantern shape in stage 8 extracts (*Figure 3E*). These experiments identify MT destabilization by kif2a as an important activity to establish the smaller stage 8 spindle architecture in embryo extracts.

## Importin α levels regulate kif2a activity and spindle recruitment to set spindle size

In the absence of changes in kif2a levels (*Figure 3C*), how is this microtubule-destabilizing factor inhibited in stage 3 extracts and recruited abundantly to stage 8 spindles? We reasoned that the observed shift in spindle assembly pathways might contribute. Regulation of spindle assembly factors by nuclear transport proteins is central to the chromatin-mediated spindle assembly pathway in meiosis and at stage 3 (*Figure 2B*), but cytoplasmic levels of importin α decrease by almost 50% at stage 8, a change

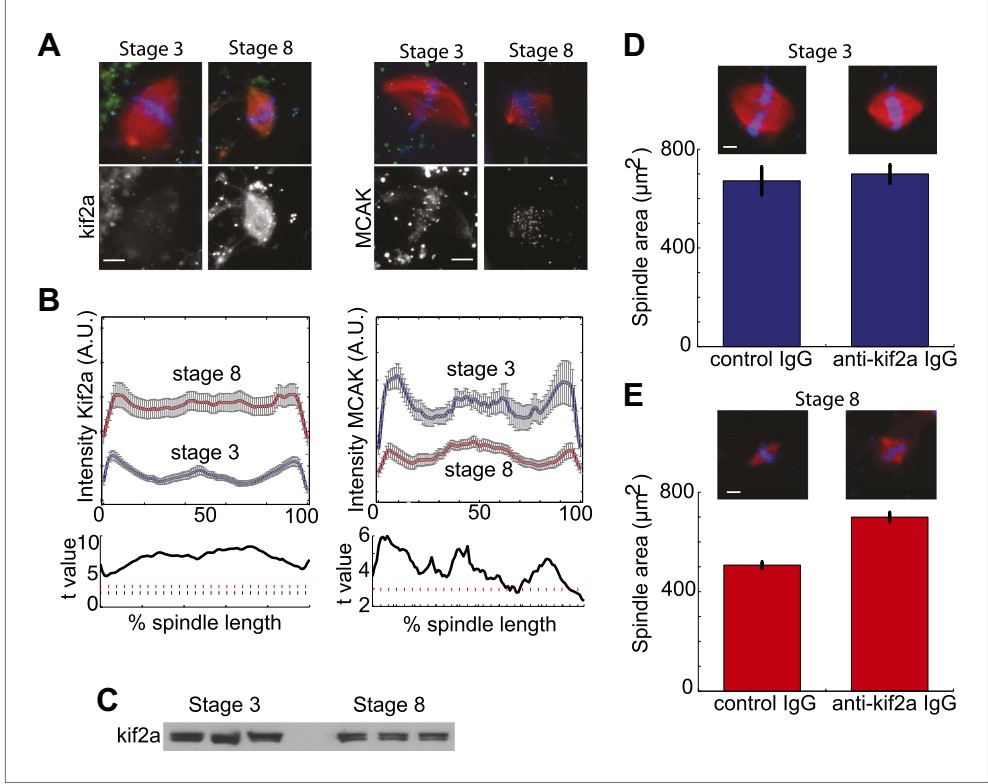

**Figure 3**. Stage 8 extract spindles recruit kif2a. (**A**) Kinesin-13s Kif2a and MCAK detected in stage 3 and stage 8 extract spindles. Top panels show microtubules (MTs) (red), chromatin (blue), and either Kif2a or MCAK as indicated (green). Bottom panels show just Kif2a or MCAK immunofluorescence within the spindle. (**B**) Upper panels: Line scan quantification of immunofluorescence intensity with normalized spindle lengths for statistical analysis. Lower panels: t-test analysis in each length bin. Black dashed line is equivalent to a p value of 0.01 and red dashed line is equivalent to a p value of 0.001. Mean ± SE from three extracts. (**C**) Immunoblot from three extracts each of stage 3 and stage 8 Kif2a indicate average levels of kif2a in each extract. (**D**) and (**E**) Inhibition of kif2a with 5 µg/ml inhibitory anti-kif2a antibody in stage 3, p=0.668 by unpaired t-test (**D**) and stage 8 p<0.0001 by unpaired t-test (**E**). Representative images are show for each condition. Mean ± SE, n ≥ 65 spindles from three extracts for each condition.

The following figure supplements are available for figure 3:

**Figure supplement 1**. Fluorescence intensity analysis of MT dynamics modulators.

that affects nuclear size during development (*Levy and Heald, 2010*), raising the possibility that importin α could impact spindle MT dynamics if it were able to regulate kif2a recruitment and activity.

Within the kif2a sequence, we identified a monopartite NLS in a surface-exposed loop of the motor domain that could potentially mediate importin α binding (*Figure 4A*), and determined that this NLS is necessary to drive nuclear import of the recombinant kif2a motor domain in interphase egg extract (*Figure 4—figure supplement 1*). An interaction between kif2a and importin α could be detected by co-immunoprecipitation (IP) from egg extracts. Since IP with kif2a antibodies could co-IP importin α, but IP with importin α antibodies did not efficiently co-IP kif2a (*Figure 4B*), this interaction appears to be weak compared to other known importin α cargos, such TPX2 that can be co-depleted with importin α (*Gruss et al., 2001*). This indicates that the binding affinity between kif2a and importin α may be in a regime that is sensitive to the levels of both proteins, as well as to cargos that compete for binding to importin α. In vitro, recombinant importin α and kif2a formed a complex detected by size exclusion chromatography, as a significant fraction of importin α shifted to a higher molecular weight and coeluted with kif2a (*Figure 4—figure supplement 2*). To test whether importin α directly regulates kif2a

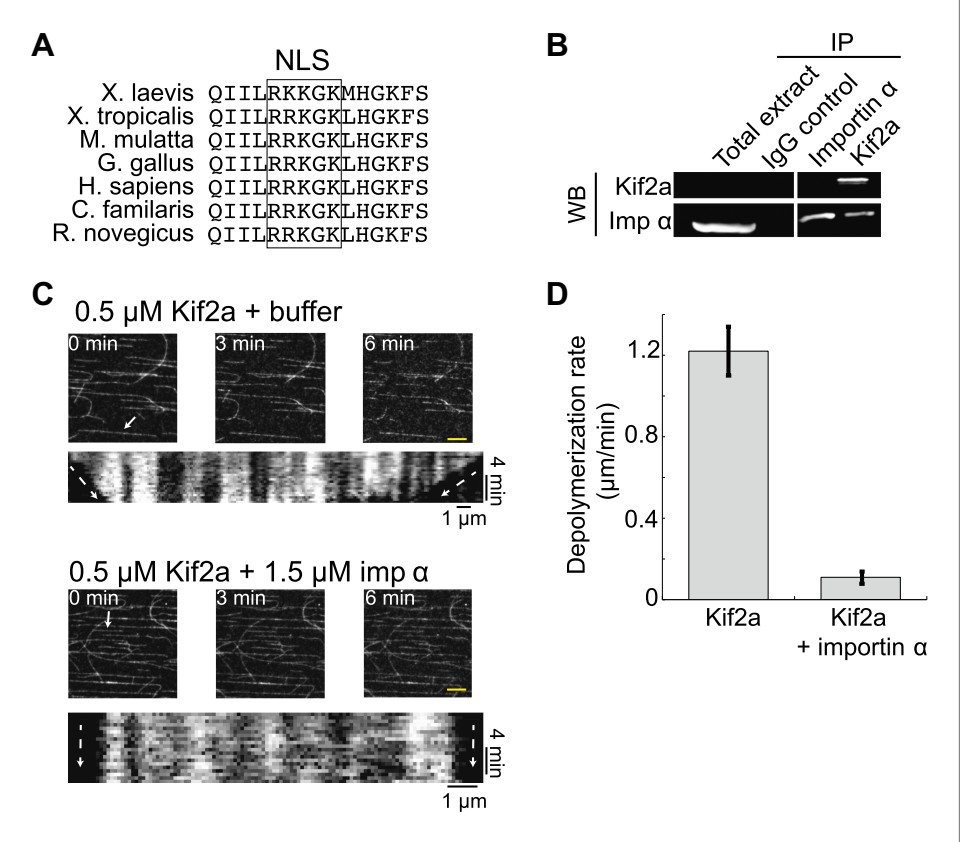

**Figure 4**. Kif2a is regulated by importin α. (**A**) Sequence alignment of region surrounding the nuclear localization signal (NLS) identified using PredictNLS program. The boxed NLS sequence is conserved within these species. (**B**) Immunoprecipitation with antibodies specific for kif2a, importin α, or control IgG from egg extract indicates an interaction between kif2a and importin α that maybe subject to strong competition by other cargos. Immunoblot with the indicated antibodies detect components of the immunoprecipitation. (**C**) Time-lapse images and kymographs from a flow cell depolymerization assay using taxol-stabilized MTs and purified full-length kif2a or kif2a plus purified importin α. Kymographs are derived from individual MTs indicated in the time 0 frame by white arrows. The scale bar represents 5 µm in time-lapse frames. (**D**) Quantification of the in vitro depolymerization rate of kif2a with and without importin α. Mean ± SE, n ≥ 20 MTs from two independent experiments, p<0.0001 by unpaired t-test.

The following figure supplements are available for figure 4:

**Figure supplement 1**. Kif2a is regulated by importin immunofluorescence detection of recombinant MBP or MBP fused to the motor domain of kif2a added at 250 nM to interphase egg extracts, showing kif2a nuclear import that is lost upon mutation of the NLS.

**Figure supplement 2**. Kif2a interacts with importin size exclusion chromatography of kif2a and importin α or importin α alone on a superdex 200 column.

**Figure supplement 3**. Kif2a is regulated by importin.

activity, the proteins were added to fluorescent taxol-stabilized MTs in a flow chamber in the presence of an ATP-regenerating system. Under these conditions, 0.5 µM kif2a depolymerized MTs at 1.2 µm/min in an ATP-dependent manner, while preincubation with 1.5 µM importin α prevented MT depolymerization (*Figure 4C,D*, *Figure 4—figure supplement 3*, and *Video 3*), indicating that importin α directly regulates kif2a.

To investigate whether the balance of kif2a and importin α could influence spindle size during development, we altered the levels of each protein in embryo extracts. Increasing the concentration

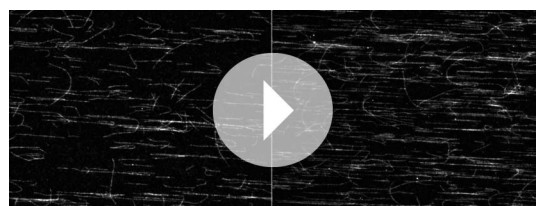

**Video 3**. In vitro depolymerization of MTs by kif2a is inhibited by importin α, related to **Figure 4**. Immobilized taxol-stabilized MTs were incubated with kif2a (500 nM) and an ATP-regenerating system or under the same condition with importin α added to threefold the kif2a concentration. Images were collected every 20 s for 5–10 min.

of kif2a in stage 3 extracts decreased spindle size in a dose-dependent manner up to ~22% (**Figure 5A**). Conversely, increasing importin α levels in stage 8 extracts lengthened spindles by ~27%, and the intensity of kif2a immunofluorescence concomitantly decreased (**Figure 5B,C**). The effects of importin α could be reversed by increasing kif2a levels, and kif2a with a mutated NLS (kif2a ΔNLS) was more effective at returning spindles to smaller sizes (**Figure 5D**). Finally, one-cell embryos injected with importin α mRNA for overexpression led to a ~20% increase in spindle length at stage 8, indicating that the interplay between importin α and kif2a can occur in developing embryos (**Figure 5—figure supplement 1**).

## Importin α is modified during development and partitions with lipid

Since the ratio of kif2a to importin α was critical for developmental spindle scaling, we investigated in more detail the mechanisms regulating importin α levels. Although previous data showed that cytoplasmic importin α levels decreased by stage 8 of *Xenopus* development (**Levy and Heald, 2010**), immunoblotting of importin α in whole-embryo lysates, rather than cytoplasmic extracts, revealed that importin α partitions into multiple populations with no apparent decrease in total amount. The single bands present stage 3 and stage 5 embryos convert to multiple bands in stage 7 and stage 8 embryos (**Figure 6A**) coincident with the initiation of spindle scaling. By examining different fractions generated by the crushing centrifugation of embryos during extract preparation, we identified a slower-migrating species of importin α enriched in the lipid/membrane fraction at stage 8, thereby decreasing importin α levels in the cytoplasmic fraction. Such a species was not detectable in whole-embryo immunoblots of stage 3 or stage 5 extracts (**Figure 6A**).

Immunofluorescence of importin α in stage 3 embryos showed a diffuse cytoplasmic localization with levels often increasing toward the animal pole, while in stage 7 embryos, where we first detect different populations of importin α by immunoblotting, the protein was enriched at the cellular boundaries (**Figure 6B**). Comparing the ratio of importin α at the cell periphery to cytoplasm by immunofluorescence intensity revealed a significant redistribution to the outer membrane at stage 7 vs stage 3. These findings point to a mechanism by which levels of cytoplasmic importin α available to regulate kif2a are sensitive to the surface area to volume ratio of plasma membrane and cytoplasm.

To examine whether the redistribution of importin α to the membrane fraction was sufficient to reduce its interaction with kif2a in the cytoplasm, we measured the amount of importin α capable of interacting with recombinant kif2a added to cytoplasmic extracts at stage 3 and stage 8 in quantitative pull-down experiments (**Figure 6C,D**). Correcting for nonspecific binding and utilizing kif2a ΔNLS as control, we found that kif2a could retrieve ~10 times more importin α from stage 3 extracts than from stage 8 extracts. This result suggests that the amount of importin α partitioning to membrane is sufficient to alter its interaction with kif2a (**Figure 6C,D**).

## Metaphase spindle orientation in large blastomeres is sensitive to spindle size

The identification of a mechanism contributing to spindle size control in embryos enabled us to test its functional significance during development. At the end of the four-cell stage (stage 3), the *X. laevis* embryo undergoes a synchronous asymmetric division with spindles aligned along the animal–vegetal axis, but displaced toward the animal pole to generate an eight-cell embryo (stage 4) containing four smaller and four larger cells. By microinjecting recombinant kif2a just prior to this division, we could specifically alter spindle size and assess both chromosome segregation and spindle orientation defects.

As in embryo extracts, excess kif2a decreased spindle size and kif2a ΔNLS possessed greater activity, reducing spindle length up to ~32% in the developing embryo (**Figure 7A**). Despite this significant size decrease, we did not observe any obvious problems with mitotic timing, metaphase chromosome

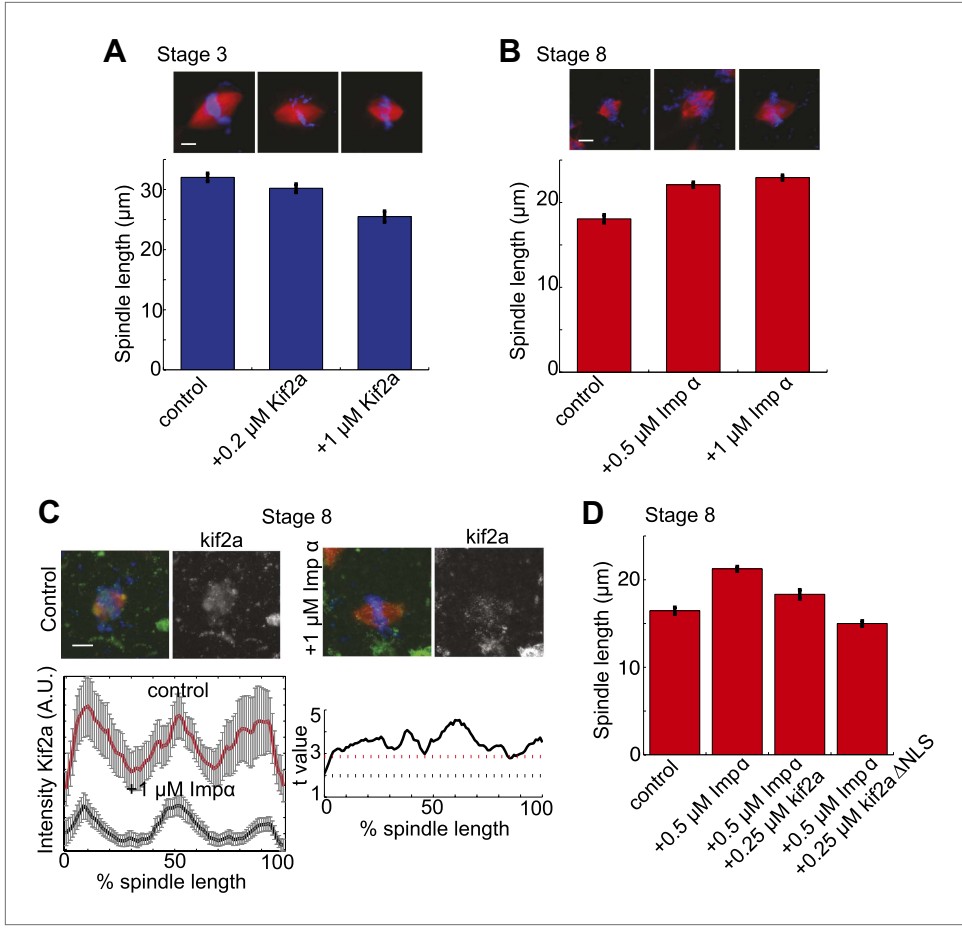

**Figure 5**. Regulation of kif2a by importin α scales spindles size. (**A**) Addition of recombinant kif2a to stage 3 extracts decreases spindle size in a dose-dependent manner. Mean ± SE, n ≥ 48 spindles from two extracts, p<0.0001 by Kruskal–Wallis test. Scale bars represent 10 μm. (**B**) Addition of recombinant importin α to stage 8 extracts increases spindle size. Mean ± SE, n ≥ 86 spindles from three extracts, p<0.0001 by Kruskal–Wallis test. (**C**) Addition of 1 μM recombinant importin α to steady-state spindles in stage 8 extracts decreases kif2a recruitment to spindles in addition to increasing spindle size. Left panel: Line scan quantification of immunofluorescence intensity with normalized spindle lengths for statistical analysis. Right panel: t-test analysis across each length bin. Black dashed line is equivalent to a p value of 0.01 and red dashed line is equivalent to a p value of 0.001. Mean ± SE, n ≥ 44 spindles from three extracts. (**D**) Addition of recombinant kif2a can rescue the increased spindle size caused by importin α addition in stage 8 extracts. An NLS mutant of kif2a (kif2a ΔNLS) is more robust at returning spindles to normal size. Mean ± SE, n ≥ 55 spindles from two extracts.

The following figure supplements are available for figure 5:

**Figure supplement 1**. Importin α overexpression increases spindle size in stage 8 embryos.

alignment, or defects during anaphase such as lagging chromosomes. These data suggest that although increasing levels of kif2a decrease spindle size, the mitotic apparatus remains architecturally functional and chromosome segregation is robust to spindle size changes, as might be expected if we were manipulating a physiological mechanism of spindle scaling. Interestingly, however, orientation of metaphase spindles was frequently randomized in the presence of excess kif2a compared to control spindles that were typically nearly aligned with the long animal–vegetal axis of the cell (*Figure 7A,B*). There was a clear correlation between metaphase spindle length and angle relative to the long axis, with shorter spindles less likely to be aligned even in unperturbed control embryos (*Figure 7C*). This effect was specific to metaphase as prometaphase and anaphase spindles were aligned along the long axis whether exogenous kif2a was injected or not (*Figure 7—figure supplement 1A*). While we

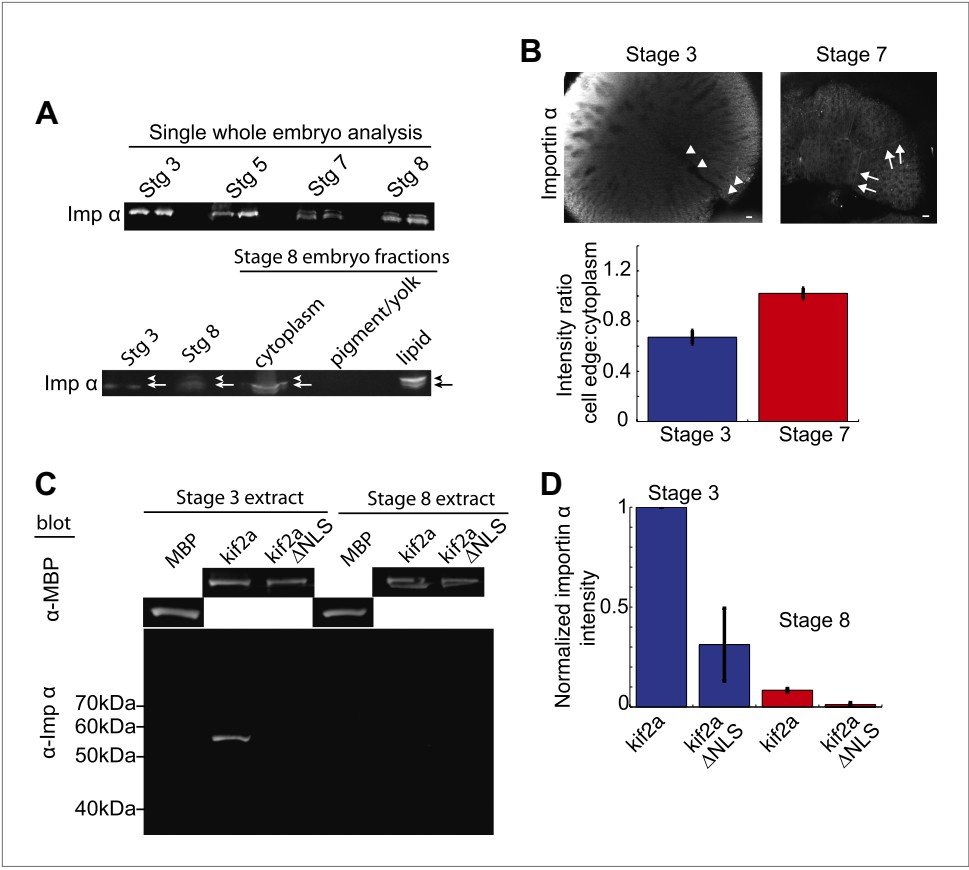

**Figure 6**. Partitioning of importin α to membrane decreases its cytoplasmic levels and binding to kif2a.
(**A**) Detection of importin α in single whole embryos (two each) at different stages of development or in different fractions from a multiple embryo extract by immunoblotting with importin α antisera. Arrows in the lower panel indicate the multiple populations of importin α that partition into different fractions. (**B**) Immunofluorescence detection of importin α in embryos at different stages and quantification of the spatial distribution by determining the relative fluorescence intensity of importin α at the cell periphery vs the cytoplasm. Mean ± SD, n = 12 and 17 cells from more than nine embryos, p <0.0001 by unpaired t-test. Scale bars represent 40 μm. (**C**) Pull down of importin α from stage 3 or stage 8 embryo extracts by recombinant kif2a. Immunoblot of MBP indicates equivalent levels of kif2a retrieved, while immunoblot of importin α shows amounts of extract importin α interacting with the MBP control, kif2a or kif2a ΔNLS. (**D**) Quantification of importin α intensity in immunoblots from pull down experiments normalized to kif2a intensity. Mean ± SE, n = 3 extracts each.

cannot rule out that orientation defects result in part from unknown pathways in which kif2a functions or from altered astral MT dynamics at metaphase, no effect was observed on prometaphase centrosome position and anaphase spindle orientation. Furthermore, spindle orientation in very large cells is established by orienting centrosomes during the previous telophase/interphase transition and not by astral MTs contacting the cell cortex hundreds of microns away (**Wuhr et al., 2011**). Finally, astral MTs were clearly evident at metaphase and early anaphase following injection of kif2a or ΔNLS kif2a (**Figure 7—figure supplement 1B**). Altogether, these data indicate that spindle length, which is controlled by regulation of MT-destabilizing activities within the spindle, impacts metaphase spindle orientation in the very large cells of the *Xenopus* embryo, but additional mechanisms operate to ensure properly positioned cleavage divisions.

## Discussion

Reconstituting spindle formation in embryo extracts enabled us to elucidate a mechanism that coordinates spindle size with cell size during *X. laevis* development through differential activation of the kinesin-13 kif2a. While a number of perturbations have been shown to alter spindle size, we have

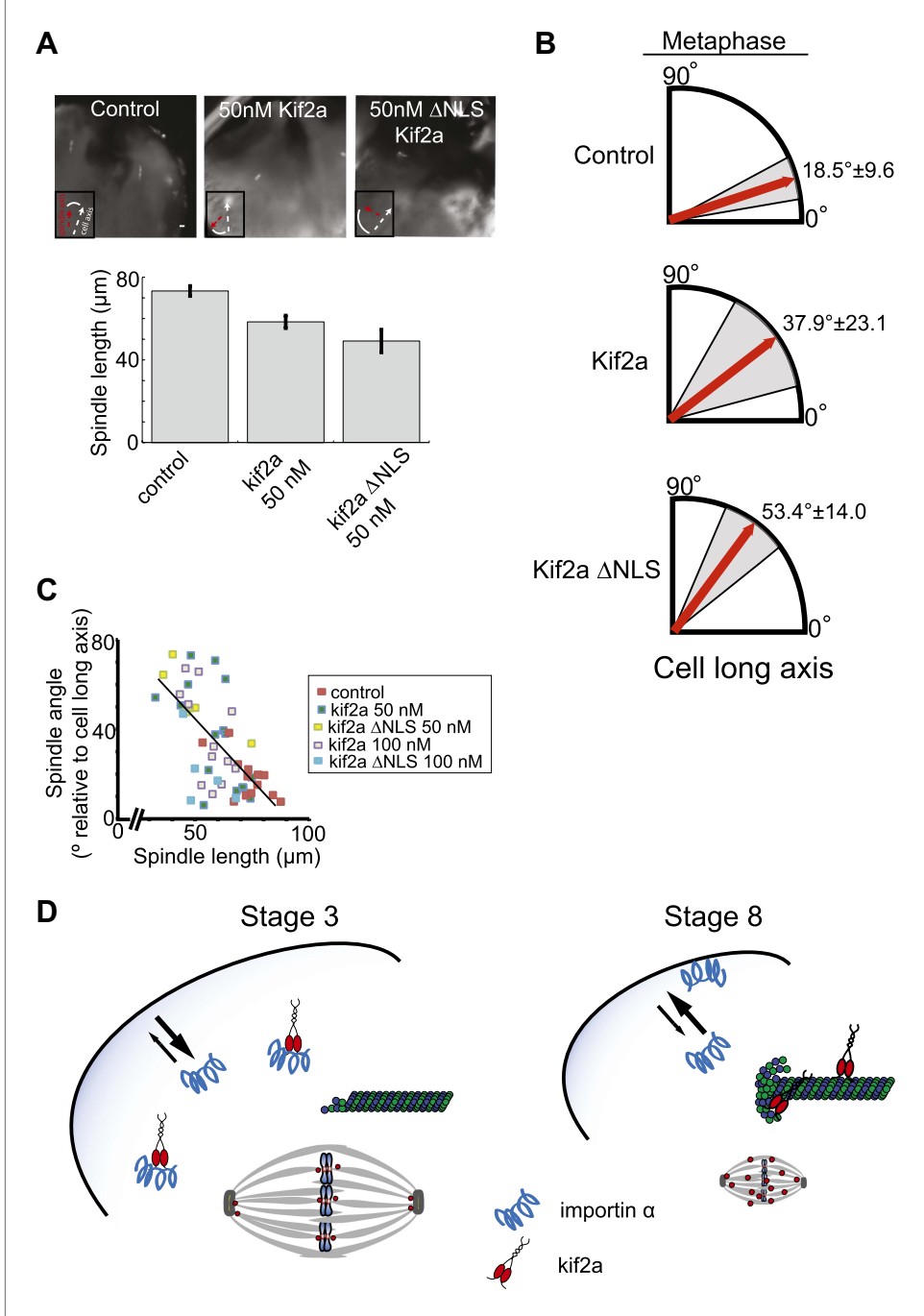

**Figure 7**. Metaphase spindle orientation in stage 3 embryos depends on spindle size. (**A**) Microinjection of kif2a into two-cell embryos just prior to cytokinesis decreases spindle size in the subsequent mitosis. Representative images of decreased spindle size in fixed hemisected embryos. Spindle orientation (pole-to-pole axis, red arrow, vs long axis of the cells, white arrow) is indicated in bottom left. The scale bar represents 40 μm. (**B**) Quantification of spindle orientation in stage 3 embryos after microinjection of kif2a or kif2a ΔNLS at metaphase. Only metaphase spindle orientation is affected. Mean ± SD, n = 6–15 spindles, p<0.01 by unpaired t-test for control vs kif2a injected and p<0.0001 for control vs kif2a ΔNLS injected. (**C**) Correlation plot of spindle length and spindle orientation in stage 3 embryos. Each point is a single spindle, and the line a linear regression of all the data. (**D**) Model of how kif2a regulation by importin α regulates spindle size during development.

The following figure supplements are available for figure 7:

**Figure supplement 1**. Prophase and anaphase spindle orientation and astral MTs.

identified, out the hundreds of known spindle associated proteins, physiological effectors that operate to scale the spindle during embryogenesis. Considering our recent work identifying the microtubule severing enzyme katanin as an activity that scales spindle sized between related frog species, modulation of MT destabilization has now emerged as a general means of physiologically scaling spindle size in *Xenopus* (*Loughlin et al., 2011*). In contrast to interspecies spindle scaling, which relied on a regulatory phosphorylation site, the developmental mechanism described here depends on decreasing cytoplasmic levels of an inhibitor, importin α, which may be directly linked to changes in the ratio of surface membrane to cell volume (*Figure 7D*). Importantly, this mechanism functions cell autonomously and independent of developmental stage, thereby assuring that the spindle scales correctly over the range of cell sizes present in the embryo at each cleavage division. It is important to note that neither the interspecies nor developmental scaling mechanisms identified fully account for spindle size differences. Future research will tell whether many factors are involved and if there are other common themes.

In principle, any pathway containing factors that partition between the cytoplasm and membrane could have altered output with changing cell size during embryogenesis. Similarly, rounds of DNA, centrosome, or other organelle replication in the absence of cell growth could provide exponentially growing sinks for inhibitory factors or drive spatial redistribution of molecules that contribute to intracellular scaling. Structural components may also become limiting as the cytoplasmic volume decreases, or other aspects of changing cell geometry could alter spatial distributions and reaction rates. It is an open question how many of these processes function to modulate the size of the spindle and other organelles, compared to the contribution of zygotic transcription and other regulatory mechanisms that turn on later in development.

We have shown that developmental spindle scaling is coincident with a change in spindle assembly pathway from a chromatin-mediated mechanism to a centrosome-dominated mechanism. In egg extracts and large blastomeres, importin α binds to cargo proteins and provides, in combination with importin β, functional regulation downstream of the RanGTP gradient (*Nachury et al., 2001*; *Wiese et al., 2001*). Although the mechanisms that regulate spindle assembly through the RanGTP gradient and downstream effectors are poorly understood (*Kalab et al., 2006*), we show that the switch in spindle assembly to a more centrosome-mediated pathway occurs concomitantly with a decrease in cytoplasmic levels of importin α. This mechanism is consistent with the gradual change from meiotic to mitotic spindle morphology observed during early mouse development and suggests that the physical changes in embryonic cells size can impact biochemical reactions that alter spindle architecture (*Courtois et al., 2012*). In egg extracts, importin α phosphorylation is thought to liberate it from membrane stores into the cytoplasm, thereby promoting its transport activity (*Hachet et al., 2004*). Whether the multiple importin α bands in stage 7 and stage 8 embryos represent different phosphorylation states or some other posttranslational modification remains to be determined. Interestingly, nuclear size is also sensitive to importin α levels, and in this way, nuclear and spindle size may be coordinated, positioning importin α as a central molecule in controlling the size of multiple subcellular structures (*Levy and Heald, 2010*).

While the RanGTP gradient is known to act through activation of MT stabilizers and motor proteins, we have identified kif2a as the first MT-destabilizing protein that could be regulated by this pathway (*Walczak and Heald, 2008*). Consistent with a role for kif2a in spindle size control, either increasing its levels or disrupting its localization by inhibiting dynein can modulate meiotic spindle size in *Xenopus* egg extracts (*Gaetz and Kapoor, 2004*; *Ohi et al., 2007*). In contrast, inhibiting the other kinesin-13 family member MCAK dramatically affects global MT dynamics (*Ohi et al., 2007*). This difference may be due to high importin α levels limiting the activity of kif2a in egg extract spindles. Whether importin α regulates kif2a in cultured cells, where it is necessary for bipolar spindle assembly and poleward flux of MTs (*Ganem and Compton, 2004*; *Ganem et al., 2005*), is unknown. Kif2a phosphorylation by aurora and polo kinases modulates its activity, which is also regulated through an interaction with inner centromere Kin-I stimulator (ICIS) (*Jang et al., 2009*; *Knowlton et al., 2009*). A comprehensive evaluation of kif2a's many interactions will help determine how layers of regulation are integrated to control developmental spindle scaling.

By identifying a physiological mechanism that controls spindle size and then altering it, we have identified a role for spindle size in maintaining spindle orientation in very large cells of the early *X. laevis* embryo. However, chromosome segregation was robust to decreases in spindle size indicating that spindle architecture remained viable and a large spindle is not essential for division of large

cells. Previous work has suggested that in contrast to the canonical spindle positioning pathways, astral MTs cannot reach the cortex of large cells to provide position cues and forces at metaphase (*Mitchison et al., 2012*). Instead, centrosomes are likely positioned prior to spindle assembly through interactions with telophase MT asters of the previous division (*Wuhr et al., 2010*, *2011*), a mechanism that does not appear disrupted under conditions that reduce spindle size in our experiments. The randomized orientation of smaller spindles specifically at metaphase in stage 3 embryos suggests that in addition to initial positioning of centrosomes to define spindle orientation, there is a mechanism that maintains spindle orientation until anaphase onset, after which large anaphase asters may again provide orientation cues. Considering the spindle as a prolate ellipsoid in a viscous medium, the frictional resistance to rotation is nonlinearly related to length. A larger spindle may therefore resist rotation, especially given the highly viscous cytoplasm and the relatively short metaphase duration in blastomeres. While identifying the mechanisms that control spindle size revealed a cellular role in spindle orientation, in the future, it will be interesting to examine how changing spindle size affects the developmental program of the embryo. Overall, our embryo extract system, in combination with biochemical validation of regulatory interactions and experimental manipulation in developing embryos, allows the identification of physiologically relevant molecular mechanisms controlling spindle architecture and cell division.

## Materials and methods

### Cytoplasmic extracts

*Xenopus laevis* egg extracts were made as previously described (*Maresca and Heald, 2006*). *Xenopus laevis* embryo extracts were generated by fertilizing eggs in vitro with crushed testes, dejellying with 3% cysteine in water pH 7.7, and allowing development to proceed at 23°C to a time point ~15 min prior to the desired stage for extracts or fully to the desired stage for embryos (*Sive et al., 2000*). For stage 3 extracts, this was 1.75 hr at room temperature, and for stage 8, this was 5.5 hr. Embryos were then washed extensively in EmCSF-XB (5 mM EGTA, 100 mM KCl, 3 mM $MgCl_2$, 0.1 mM $CaCl_2$, 50 mM sucrose, and 10 mM HEPES pH 7.7) plus protease inhibitors (leupeptin, pepstatin, and chymostatin, 10 µg/ml each) and cytochalasin D (20 µg/ml), packed by centrifugation in a 2-ml microfuge tube at 1000×$g$ for 1 min followed by 2000×$g$ for 10 s. All buffer was removed, and embryos were crushed by centrifugation at 17,000×$g$ in a swinging bucket rotor (Sorvall HB-6, Thermo Fisher Scientific, Waltham MA, USA). Concentrated cytoplasm was removed from the tube and immediately placed on ice. Protease inhibitors, cytochlasin D, and energy mix were added as for egg extracts and UbcH10 C114S was added at 0.2 mg/ml final concentration. Cyclin B delta 90 (0.05 mg/ml) was added at the start of spindle reactions or prior to utilizing the extract for biochemical analysis. Purified stage 8 nuclei were generated by pelleting endogenous nuclei from stage 8 extracts multiple times through ELB buffer (250 mM sucrose, 50 mM KCl, 2.5 mM $MgCl_2$, and 10 mM HEPES pH 7.8).

### Immunofluorescence analysis

The antibody to kif2a raised against the N-terminal domain was a gift from C. Walczak. Importin α antisera were raised against full-length *Xenopus* importin α (*Levy and Heald, 2010*). Anti-MBP monoclonal antibody is commercially available from New England Biolabs. Immunofluorescence analysis was performed as previously described (*Loughlin et al., 2011*). Briefly, spindles in extract were fixed with 3.7% formaldehyde, sedimented through a 40% glycerol cushion onto coverslips, post-fixed in 100% methanol, and blocked with PBS – 1% BSA. Coverslips were incubated to equilibrium with 1:1000 dilution of primary antibody, washed extensively, and incubated with a 1:1000 dilution of secondary antibody (Alexa-labeled anti-rabbit or anti-mouse; Invitrogen).

Image collection was as previously described (*Loughlin et al., 2011*) except the images in *Figure 5F* and *Figure 7—figure supplement 1* utilized the same imaging system with an addition of a patterned illumination optigrid device (Qioptic, Rochester NY, USA) to decrease background autofluorescence in embryos. For quantification of immunofluorescence, photobleaching tests were performed on extra samples to identify the total exposure time a field could be illuminated without significantly changing the measured fluorescence intensity, and image collection was maintained within this time window. Lamp intensity variation was estimated by imaging of tetraspeck beads (Invitrogen/Life Technologies, Grand Island NY, USA) and maintained to within ~10% as judged by bead emission intensity.

Immunofluorescence quantification was performed by manually collecting 25-pixel-width linescans in ImageJ, locally correcting for background intensity if it was more that 10 intensity units above the camera offset. Data was then normalized to 100% spindle length, and intensity was averaged within each 1% length bin. Depending on the length of the spindle, ~2 to 6 pixels were averaged per length bin. Spindles from each condition were averaged under these normalized spindle length conditions with error propagation and intensity plotted as a function of 0–100% spindle length. The t-test analysis was then used when comparing the significance of any differences within each 1% length bin, and the t value was plotted for each bin. The t values for each dataset with statistically significant p values of <0.01 are indicated by a black dashed lines and <0.001 are indicated by red dashed lines.

Immunofluorescence intensity ratios of importin α in embryos were determined by calculating the integrated densities of a 6-pixel-width line along the cell periphery compared to the same line shifted ~20 pixels into the cytoplasm of the cell.

## Protein purification

Recombinant importin α utilized for all extract and embryo experiments was a multiple phosphomimetic mutant proficient for nuclear import in egg extracts (*Levy and Heald, 2010*). Recombinant importin α used for in vitro biochemical experiments had the first 43 amino acids deleted to prevent known autoinhibitory interactions, but did not contain phosphomimetic mutations. Full-length kif2a and motor domain (amino acids 204–538) were amplified by polymerase chain reaction, cloned in to pMAL-C5X (New England Biolabs, Ipswitch MA, USA), and expressed and purified as described for MBP fusion proteins (*Loughlin et al., 2011*). The NLS mutant had three charge switch mutations, K427E, K428E, and K430E, introduced into the NLS sequence identified by Quikchange mutagenesis (Stratagene/Agilent Technologies, Santa Clara CA, USA).

## Pull down experiments and immunoblotting

1 μM recombinant kif2a or kif2a ΔNLS as an MBP fusion protein or MBP alone was added to extracts and incubated for 15 min. 75 μl of Amylose resin (New England Biolabs) was then incubated for 15 min to retrieved MBP or fusion proteins. Resin was washed with a total of 25 bed volumes of phosphate-buffered saline with 0.01% Tween and eluted in SDS-PAGE loading buffer. Immunoblotting utilized 1:1000 dilution of all antibodies or antisera and was performed as described (*Loughlin et al., 2011*).

## Microinjection into embryos

Embryos were injected at either the one-cell stage with in vitro transcribed RNA coding for importin α or into one blastomere at the two-cell stage with recombinant protein in injection buffer (10 mM potassium glutamate, 1 mM $MgCl_2$). A homemade pneumatic injection apparatus was utilized with microcapillary needles and calibrated by injecting dye into mineral oil. The total volume injected was approximately 10 nl.

## Acknowledgements

We thank members of the Heald laboratory for helpful discussions, and Karsten Weis for critical comments on the manuscript. Special thanks to Claire Walczak for Kif2a and MCAK antibodies, and to Ron Vale for patronin antibodies.

# Additional information

## Funding

| Funder | Grant reference number | Author |
| --- | --- | --- |
| National Institutes of Health | R01GM098766 | Rebecca Heald |

The funder had no role in study design, data collection and interpretation, or the decision to submit the work for publication.

## Author contributions

JDW, Conception and design, Acquisition of data, Analysis and interpretation of data, Drafting or revising the article; RH, Conception and design, Analysis and interpretation of data, Drafting or revising the article

## Ethics

Animal experimentation: This study was performed in strict accordance with the recommendations in the Guide for the Care and Use of Laboratory Animals of the National Institutes of Health. All the animals were handled according to approved institutional animal care and use committee (IACUC) protocols of the University of California. The protocol was approved by the Animal Care and Use Committee (Permit Number: R238-1112). Every effort was made to minimize suffering.

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
