## [Decision Letter]

Thank you for choosing to send your work entitled “Mitotic spindle scaling during *Xenopus* development by kif2a and importin α” for consideration at *eLife*. Your article has been evaluated by a Senior editor (Detlef Weigel) and 2 reviewers, one of whom is a member of our Board of Reviewing Editors.

The Reviewing editor and the other reviewers discussed their comments before we reached this decision, and the Reviewing editor has assembled the following comments based on the reviewers' reports.

Your manuscript has been read and reviewed by two experts, who agree that your work is of general interest as scaling of cellular structures is a topic of foremost importance in modern cell biology. Both reviewers make the point that the first part of the paper (extract systems and the implications for changes in the spindle throughout development) is profound and very important, while the data presented in the second part (kif2a and importin α are responsible for the difference between spindle length) are not strong enough to provide extensive, mechanistic insights. However, we do believe that your work is worth publishing but for resubmission you would have to improve the second part as proposed below.

A major claim in the paper is that at stage 8, spindle size is reduced upon release of an inhibitory interaction between the kinesin kif2a and importin α. This occurs through redistribution of importin α from the cytoplasm to a membrane fraction. This point needs to be shown in a more direct and quantitative manner. We would ask you to measure the cytoplasmic levels of active importin α, data that can be collected either with (1) FCS, (2) FRET (sensors described in Sun et al., 2007, PNAS), or (3) quantitative pull downs. We suggest this because we expect that the reagents already exist in your lab, based on previous papers you have published.

---

## [Author Response]

*A major claim in the paper is that at stage 8, spindle size is reduced upon release of an inhibitory interaction between the kinesin kif2a and importin α. This occurs through redistribution of importin α from the cytoplasm to a membrane fraction. This point needs to be shown in a more direct and quantitative manner. We would ask you to measure the cytoplasmic levels of active importin α, data that can be collected either with (1) FCS, (2) FRET (sensors described in Sun et al., 2007, PNAS), or (3) quantitative pull downs. We suggest this because we expect that the reagents already exist in your lab, based on previous papers you have published*.

We have addressed the concern with a quantitative pull down experiment. Tagged recombinant kif2a was added to stage 3 or stage 8 extracts and then retrieved, revealing the amount of active importin α in the each extract that is free to bind kif2a, using a kif2a NLS mutant as control (Figure 6). We found that in stage 3 extracts importin α could bind kif2a while in stage 8 extracts approximately ten-fold less importin α interacted with the added kif2a (quantified in Figure 6).

In addition, we attempted to further quantify the kif2a-importin α by other approaches suggested by the reviewers. FCS experiments were performed upon addition of fluorescently labeled kif2a motor domain to either stage 3 or stage 8 extract at low concentrations. Fluorescence intensity was measured at 500 frames/sec and processed as by the PACI method developed by Needleman et al. (Biophysics J. 2009). Correlation curves indicated slower diffusion of kif2a motor domain in stage 3 compared to stage 8 extracts, indicating the interaction with other proteins and consistent with the formation of importin α-containing complexes. However, curve fits to this data were underdetermined for some parameters and therefore, despite qualitatively supporting our pulldown results, we feel the data are not strong enough to warrant addition to the manuscript.

Experiments utilizing FRET or other methods that require a fluorescently labeled version of importin α would require addition or overexpression of importin α and therefore perturbation of its levels. Given the small changes in importin α partitioning that affected spindle size, we were unable to interpret data from the system perturbed in this way since any additional importin α protein is expected to affect its steady state distribution.

Overall, the addition of the quantitative pulldown data strongly supports our model that partitioning of importin α out of the cytoplasm can decrease its interaction with kif2a, leading to a change in spindle size.